# Novel Apoptosis-Inducing Agents for the Treatment of Cancer, a New Arsenal in the Toolbox

**DOI:** 10.3390/cancers11081087

**Published:** 2019-07-31

**Authors:** Bora Lim, Yoshimi Greer, Stanley Lipkowitz, Naoko Takebe

**Affiliations:** 1Department of Breast Medical Oncology, The University of Texas MD Anderson Cancer Center, Houston, TX 77030, USA; 2Women’s Malignancies Branch, Center for Cancer Research, National Cancer Institute, Bethesda, MD 20892, USA; 3Early Clinical Trials Development, Division of Cancer Treatment and Diagnosis, National Cancer Institute, Bethesda, MD 20892, USA

**Keywords:** apoptosis, cancer, intrinsic apoptosis, extrinsic apoptosis, biomarker, targeted therapy

## Abstract

Evasion from apoptosis is an important hallmark of cancer cells. Alterations of apoptosis pathways are especially critical as they confer resistance to conventional anti-cancer therapeutics, e.g., chemotherapy, radiotherapy, and targeted therapeutics. Thus, successful induction of apoptosis using novel therapeutics may be a key strategy for preventing recurrence and metastasis. Inhibitors of anti-apoptotic molecules and enhancers of pro-apoptotic molecules are being actively developed for hematologic malignancies and solid tumors in particular over the last decade. However, due to the complicated apoptosis process caused by a multifaceted connection with cross-talk pathways, protein–protein interaction, and diverse resistance mechanisms, drug development within the category has been extremely challenging. Careful design and development of clinical trials incorporating predictive biomarkers along with novel apoptosis-inducing agents based on rational combination strategies are needed to ensure the successful development of these molecules. Here, we review the landscape of currently available direct apoptosis-targeting agents in clinical development for cancer treatment and update the related biomarker advancement to detect and validate the efficacy of apoptosis-targeted therapies, along with strategies to combine them with other agents.

## 1. Introduction

Apoptosis is one of the established mechanisms of cancer cell death, such as necroptic cell death, lysosome-dependent cell death, necroptosis, and ferroptosis. As per the most recent Nomenclature Committee on Cell Death (NCCD) in 2018, there are 10 major established regulated cell death mechanisms despite the overlap observed among different mechanisms [1]. Apoptosis is a physiological process that plays a role in development and homeostasis by eliminating unnecessary or abnormal cells in metazoans [2,3]. An acquired resistance to apoptotic cell death is one of the hallmarks of cancer, largely due to an overexpression of anti-apoptotic genes, and downregulation or mutation of pro-apoptotic genes [4]. Therefore, overcoming resistance to apoptosis by activating apoptosis pathways has been a major focus in the development of therapeutic strategies for cancer treatment. There are two major apoptotic pathways: The death receptor-mediated pathway (extrinsic pathway) and the mitochondrial-mediated pathway (intrinsic pathway) (Figure 1).

The extrinsic pathway is induced by tumor necrosis factor (TNF) superfamily members, such as TNF, FAS, and TNF-related apoptosis inducing ligand (TRAIL) binding to death receptors (DR) (e.g., TNFR1, FAS receptor (FASR), and DR 4/5) on the cell surface followed by caspase-8/10 activation [5]. This leads to downstream caspases-3/7 activation and ultimately apoptosis [5]. Of these TNF ligands, TRAIL is an attractive therapeutic candidate because it activates the extrinsic death pathway in cancer cells with little cytotoxicity in normal cells [5], and induces apoptosis regardless of the status of the p53 tumor suppressor [6]. These ideal therapeutic characteristics of TRAIL led to the development of many DR agonists (Table 1, Appendix A).

However, TRIAL and DR agonists also have shown several challenges in therapeutic development as we will highlight below. In the cell-intrinsic pathway, when cells are stressed by internal lethal stimuli, BH3 interacting-domain death agonist (BID), Bcl-2-associated X protein (BAX), Bcl-2 homologous antagonist killer (BAK), and lipids cooperate to form supramolecular openings in the outer mitochondrial membrane, which is called mitochondrial outer membrane permeabilization (MOMP) [7], that results in cytochrome c release and apoptosome formation, leading to the apoptotic death of cells. The activators of apoptosis beyond the DR and mitochondria, such as heat shock proteins (HSPs) and endoplasmic reticulum (ER) stress inducing agents by activation of caspase 12, are also potential targets for novel therapeutics. For the interest of this specific manuscript, we will not discuss these agents, however, readers can find more information in this review [8,9,10,11].

Complex communication among various members of apoptotic regulators within and outside the pathway brings challenges to the effective induction of apoptosis in solid tumors. However, advances in genomic assays, medicinal chemistry, structural biology, and bioinformatics offer critical technologies to move the field forward. For example, the determination of the crystal structure of the B cell lymphoma 2 (BCL-2) family binding groove laid the groundwork for the development of direct intrinsic apoptosis pathway-targeting agents [12]. In addition, the application of nuclear magnetic resonance (NMR) and structure–activity relationship (SAR)-based approaches have led to the elucidation of other members of the cascade, e.g., X-Linked Inhibitor of Apoptosis (XIAP), BCL-XL, and Cellular Inhibitor of Apoptosis Protein (cIAPs), allowing the development of specific inhibitors [13,14]. More biomarker assays are also becoming available. Therefore, apoptosis, as a critical hallmark of cancer biology, continues to offer intriguing opportunities as a therapeutic target [15,16]. Further efforts to develop the category of drugs to directly target this mechanism are therefore rational and necessary. Thus, in this paper, we review the currently available agents that directly target proteins within the apoptosis pathway and briefly discuss potential partners to induce synergy, and emerging biomarker assays.

## 2. Extrinsic Pathway Targeting

### 2.1. TRAIL and Its Receptors

TRAIL (Apo-2L; gene symbol TNFSF10) is one of several members of the TNF gene superfamily, and potently induces apoptosis in its soluble and membrane-bound forms [12,13,17]. Active TRAIL is comprised of a trimer formed by three TRAIL monomers with a zinc ion buried at the trimer interface. The zinc is required for maintaining the structure and biological activity of the protein [14]. TRAIL mRNA is widely expressed in many tissue types [12,17]. TRAIL binds to five different receptors, including DR4, DR5, decoy receptor (DcR) 1, DcR2, and osteoprotegerin (OPG) (Figure 1) [15,16,18,19,20,21,22,23,24,25,26]. Activation of DR4/5 induces apoptosis, while DcR1, DcR2, and OPG lack the cytoplasmic death domain (DD) and do not induce apoptosis. TRAIL-DcR2 binding activates NF-κB [15,27] and further inhibits DR4/5-mediated apoptosis. The role of DcRs remains unclear [28]; elevated expression of DcRs in multiple cancer types are reported [29], but expression of the DcRs lacks correlation with TRAIL sensitivity in preclinical models [30]. TRAIL signaling has also been associated with non-apoptotic, caspase-independent cell death, such as necrosis and necroptosis [31,32,33,34,35].

In addition, TRAIL plays important roles in innate and adaptive immune cells [36,37,38]. Stimulated natural killer (NK), T, and dendritic cells show upregulated TRAIL [36,37]. TRAIL upregulated in response to interferon (IFN)-γ is critical for NK cell function [37]. TRAIL blockade induces thymocyte apoptosis and increased autoimmune responses [39,40,41] and promotes tumor metastasis [42] in mice. These findings support that TRAIL provides critical immune surveillance and regulatory functions, including the suppression of autoimmunity and inhibiting tumor growth and metastasis [43].

The canonical role of the TRAIL/DR pathway is to initiate caspase activation leading to apoptosis [44,45,46]. Preclinical evaluation of TRAIL in various models has shown activity in a wide range of tumor types causing regression of established tumors [44,45,46]. Preclinical work has also shown antimetastatic activity of the TRAIL/DR pathway. In mice lacking their death-inducing TRAIL receptor (mice only have one death-inducing TRAIL receptor), increased metastases in the lymph nodes were observed in a squamous cell carcinoma model [47]. Consistent with this, RNAi mediated knockdown of DR5 in human lung cancer cells increased metastases to the lungs from tumors implanted subcutaneously [48]. Experimental metastases models using injected human cancer cells into immunodeficient mice have also demonstrated that TRAIL decreases metastases to the lungs [49,50]. Whether the anti-tumor effects of TRAIL are mediated through DR4 or DR5 appears to differ depending on the model [49,51,52,53].

Paradoxically, TRAIL/DR stimulates cell growth and metastases in certain preclinical models; TRAIL promoted proliferation of pre-activated T cells [54] and cancer metastasis [55,56]. Recent data found that TRAIL can promote cytokine secretion that suppresses anti-tumor immune responses and promotes metastasis [57,58,59]. The relative role of DR4 and DR5 in stimulating metastases is unclear. One publication found that DR5, but not DR4, is able to stimulate cell migration and invasion [52]. Further characterization of the conditions under which TRAIL signaling is anti- or pro-tumor and the contributions of DR4 and DR5 to each effect is necessary to establish appropriate therapeutic strategies.

### 2.2. Clinical Development of Death Receptor (DR) Targeting Therapies

Targeting DR has been of interest for cancer therapy because of the specificity in cancer cells [12,60]. Notably, soluble recombinant TRAIL demonstrated significant tumor regression without systemic toxicity in animal models [61,62], and this led to clinical development of DR-targeted biologics, such as DR4/5 specific targeted therapy (Table 1, Appendix A).

### 2.3. First Generation

In the past decade, recombinant human TRAIL (rhTRAIL) and multiple DR agonistic antibodies were tested in clinical trials. Unfortunately, the results were disappointing despite evidence of efficacy in a limited number of cases [63,64,65,66,67,68,69,70]. Dulanermin (rhTRAIL) [62,71,72,73,74,75] did not show clear clinical responses, possibly due to its short half-life (<60 min). DR4-specific agonist mapatumumab [63,76,77,78,79,80,81,82,83], DR5-specific agonists drozitumab [64,65,84], conatumumab [66,67,85], lexatumumab [68,86,87,88,89], HGS-TR2J [90,91], tigatuzumab [69,92,93,94,95,96], and LBY135 [70] were all discontinued after phase 1 or 2 trials without impactful benefits despite their safety (Appendix A). The reasons why compelling pre-clinical findings with these agonist antibodies did not translate into more robust clinical efficacy remain poorly characterized. One possible reason may be that DR agonist antibodies require clustering via their Fc domains for effective receptor activation of apoptosis [85,97,98,99], which can be easily achieved by the addition of exogenous cross-linking agents in vitro. In vivo, interaction between the Fc portion of the agonist antibody and Fcγ receptors (FcγR) on immune cells is key to promoting tumor cell destruction [85,97,98,99]. In the clinical setting, on the other hand, expression of inhibitory FcγR or altered levels of FcγR due to polymorphisms in immune cells may hamper antibody cross-linking and suppress the anti-tumor response [99].

### 2.4. Second Generation

New DR-targeting drugs (Table 1) show improved bioactivity. 

ABBV-621/APG880 is a novel DR agonist designed to maximize receptor clustering without Fcγ-R-mediated crosslinking. ABBV-621 is comprised of the single chain TRAIL-receptor binding domain (RBD) linked to a human IgG1-Fc domain [100]. The monomers are covalently connected by glycosylated linkers, resulting in two sets of trimeric RBDs. ABBV-621/APG880 is currently being tested in a phase 1 study in solid tumors and hematologic malignancies (NCT03082209).

APG350 is a fusion protein monomer and has an extended half-life (~28 h in mouse and monkey). APG350 induces superior clustering of DRs and shows antitumor efficacy independent of cross-linking via FcγR ^45^. Preclinical studies reported promising therapeutic activity of APG350 in xenograft models of pancreatic cancer [101,102]; APG350 significantly reduced tumor burden in palliative treatment, and limited recurrent tumor growth and metastases in adjuvant therapy [102]. At present, no further development of APG350 is reported [49].

RG7386/RO6874813 is a novel bispecific antibody that binds to fibroblast activation protein (FAP) and DR5. FAP-driven binding of RO6874813 mediates high levels of DR5 clustering [103]. Preclinical studies confirmed strong apoptosis induction in tumors [104]. In a phase 1 study, RO6874813 demonstrated a favorable safety profile in patients with multiple solid tumor types and antitumor activity was observed in a patient with non-small cell lung cancer (NSCLC) [105]. As of August 2018, however, the development of RG7386/RO6874813 in solid tumors has been discontinued [106].

TAS266 is a high potency DR5 agonist tetravalent Nanobody^®^ [107], and was discontinued due to hepatotoxicity in the phase 1 study [107].

MEDI3039 is a unique multivalent high potency DR5 agonist [108,109,110]. The multivalent scaffold protein is an engineered protein based on the third fibronectin type III domain of the human glycoprotein tenascin C [109,111], which possesses a region similar to the variable region characteristic of antibodies. MEDI3039 was found to be one to two orders of magnitude more potent than TRAIL, and enhanced apoptosis was observed with increased valency of the super-agonist [49,109,112]. The effectiveness of this super-agonist has yet to undergo testing in clinical trials.

Hexabodies are novel modified antibodies that form complement-mediated hexamers on cell surfaces after antigen binding and direct complement dependent cell death [113]. *HexaBody^®^-DR5/DR5 (GEN1029)* is a mixture of two non-competing HexaBody molecules that target two distinct epitopes on DR5 [114]. A phase I/II study in solid cancers is ongoing.

CPT (circularly permuted TRAIL) is a recombinant human mutant of Apo2L/TRAIL developed by Beijing Sunbio Biotech, Co. Ltd. in China, and was tested in relapsed and refractory multiple myeloma as a single agent [115,116], or in combination with thalidomide (T), or in combination with thalidomide and dexamethasone (TD) [117,118]. Median PFS was 6.7 months in the CPT + TD group compared with 3.1 months for the TD group [118]. A phase 3 study is currently under way (ChiCTR-IPR-15006024, http://www.chictr.org.cn/).

### 2.5. Third Generation

The small molecule ONC201 (*a.k.a.* TIC10, NSC350625) was identified in a chemical library screen as an inducer of TRAIL expression in a colon cancer cell line [119,120]. ONC201 inhibits Akt and MEK activity, resulting in de-phosphorylation of the Foxo3a transcription factor and subsequent transcriptional activation of TRAIL [119,120,121]. Multiple preclinical studies reported cytotoxic effects of ONC201 in solid and hematological cancers, although the contribution of TRAIL induction is not consistent as other mechanisms of activity have been described [119,120,122,123,124,125,126,127,128]. ONC201 synergizes with chemotherapeutic and targeted agents, including sorafenib and cytarabine, in multiple preclinical models [122,123]. The oral availability and ability to cross the blood brain barrier confer ideal characteristics to ONC201 for cancer treatment [119,129]. A phase 1 study showed that ONC201 was well tolerated, achieved micromolar plasma concentrations, and was biologically active in advanced cancer patients [130]. In a phase 2 study in recurrent glioblastoma, ONC201 showed single agent activity; progression free survival (PFS) at 6 months was 12%, and one patient exhibited remarkable tumor regression [131]. ONC201 is currently being tested in multiple cancer types (Table 1).

### 2.6. Other Recent Developments

*Bioymifi* was recently identified as a small molecule that can activate DR5 as a single agent and leads to apoptosis [132]. However, no further information about preclinical development has been reported.

#### 2.6.1. Mesenchymal Stem Cell-Mediated TRAIL Delivery

Mesenchymal stem cells (MSCs) that have been engineered to express TRAIL have been explored as TRAIL delivery agents [133,134]. Because MSCs possess tumor-homing capabilities and are able to evade elimination by the immune system, they have been explored as cancer therapy delivery systems. MSCs engineered to express TRAIL have been found to induce apoptosis in multiple cancer types in vitro and in vivo [135,136,137,138,139,140] with higher potency than soluble TRAIL [140]. Cisplatin sensitized mouse glioblastoma tumors to stem cell-delivered TRAIL in vitro and in vivo [139], suggesting that combinatorial therapies effectively sensitize cancer cells to stem cell-delivered TRAIL. However, stem cell delivery systems have not yet been tested in clinical trials in cancers [134]. The potential stem cell tumorigenicity [119,133] and absence of available safety measures are concerns that need to be addressed prior to clinical translation [134]. 

#### 2.6.2. Nano Particle-Based Drug Delivery

Recent advances in drug delivery, materials science, and nanotechnology are now being exploited to develop next-generation nanoparticle platforms to improve TRAIL therapeutic delivery (reviewed in [141,142,143]). The nano-delivery technology offers the potential to improve the stability of TRAIL and prolong its half-life in plasma, to specifically deliver TRAIL to a particular target site, and to overcome resistance to TRAIL.

#### 2.6.3. CRISPR-Based TRAIL Therapy

Taking advantage of a tumor’s “self-homing” behavior, a recent study showed CRISPR-engineered self-targeting tumor cells that secrete DR ligands effectively killed the primary and metastatic tumor but did not destroy themselves [144], suggesting clinical development of cancer therapy using genome-editing. Safety issues, similar to the concerns for mesenchymal stem cells, will need to be addressed for the use of modified tumor cells for TRAIL delivery.

## 3. Challenges and Strategies to Improve the Efficacy of DR-Targeted Therapy

### 3.1. Evaluation of Pharmacokinetic and Pharmacodynamic Characteristics of DR Targeting 

The half-life of Dulanermin [62] is very short (<60 min) [66,71,72,75] and this may partially explain the observed lack of effectiveness. The DR agonist antibodies exhibited considerably longer half-lives (10 days to weeks) [64,66,77,78,87,88,93,145], however, they had little activity. Thus, a longer half-life alone may not address all of the issues that lead to poor clinical activity, but also pharmacodynamic processing of absorbed therapeutics contribute to the final efficacy as well. Apoptotic cells are rapidly engulfed and destroyed by phagocytic cells in the surrounding microenvironment [146], requiring rapid pharmacodynamic measures. Circulating DNA, active caspase-3/8, and cytokeratin 18 [147] are commonly utilized and were also used in DR antibody clinical trials [66,71,74,81,146,148]. However, the levels of these biomarkers varied among tumors and did not correlate with response. Alternatively, the M30 and M65 sandwich enzyme-linked immunosorbent assay (ELISA) assay systems are useful to evaluate cytokeratin 18 in serum [149], allowing detection of apoptosis in real time. The development of pharmacodynamic markers in parallel to new TRAIL receptor agonists will help evaluate the on-target activity of these drugs. 

### 3.2. Resistance Mechanisms to TRAIL-Induced Apoptosis

There are a number of potential mechanisms of resistance to DR agonists. The intrinsic apoptotic members, Cellular FLICE (FADD-like IL-1β-converting enzyme)-inhibitory protein (c- FLIP), inhibitor of apoptosis proteins (IAPs), and anti-apoptotic members of the BCL-2 family (BCL-XL and BCL-2) serve as negative regulators of the apoptotic program, although their expression levels are not correlated with sensitivity to TRAIL [44,51]. Elevated clathrin-mediated endocytosis of the DRs has also been proposed as a mechanism of resistance by decreasing the cell surface levels of DR4/5 [150]. However, others have described that impaired endocytosis of DRs is associated with TRAIL resistance [151]. Expression of specific microRNAs is also proposed to confer TRAIL resistance [152,153]. TRAIL decoy receptors (DcR1/2) in tumors may inhibit DR4/5-mediated apoptotic signal [154]. An evaluation of these mechanisms of resistance in clinical trials is warranted.

### 3.3. Combination Strategies 

Combining TRAIL and DR targeted agents with targeted therapies (e.g., epidermal growth factor receptor (EGFR) inhibitors, poly ADP ribose polymerase (PARP) inhibitors) has been proposed to overcome resistance to DR-targeted therapies by reducing variability among cells with respect to timing and susceptibility to an apoptosis inducer [155,156]. Moreover, DR agonists activate apoptosis independently of p53 status, providing another rationale for using DR agonists in combination with other therapeutic agents to overcome resistance to cell death [157]. Various chemotherapeutic drugs have been found in preclinical models to synergize with DR agonists, including 5FU [48,158,159], gemcitabine [160,161], irinotecan [159,162], Adriamycin [163], and paclitaxel [163]. The proposed mechanisms to sensitize cells to TRAIL include enhanced DISC formation, and up- or down-regulation of pro- or anti-apoptotic regulators. However, the results from clinical trials to date do not demonstrate that the addition of a TRAIL DR agonist to a chemotherapeutic regimen improves patient outcomes [46,164]. Alternatively, a number of targeted drugs have been shown to synergize with DR targeted therapies. Inhibition of anti-apoptotic components of the intrinsic pathway (e.g., the BCL-2 family inhibitor ABT-199 and IAP inhibiting second mitochondria-derived activator of caspases (SMAC) mimetics were shown to sensitize cells to TRAIL in a number of preclinical models [165,166,167,168,169,170,171,172]. Multiple receptor tyrosine kinase inhibitors were shown to enhance TRAIL-induced apoptosis [51,173,174]. Histone deacetylase inhibitors [175,176,177] and cell cycle regulators, such as CDK9 and WEE1 inhibitors, also sensitize cells to TRAIL [178,179]. The proteasome inhibitor, Bortezomib, is one of the most potent DR sensitizers in preclinical studies [180]. Bortezomib stimulated apoptosis by increasing expression of DRs, reduced c-FLIP and enhanced caspase 8 activation, modulated BCL-2 family proteins, and decreased expression of inhibitors of IAPs [180].

Sequential or combination treatment with radiation followed by TRAIL agents has been found to enhance apoptosis in prostate [181], breast [182], lung, colorectal, head and neck cancers [181,182,183,184,185,186,187,188,189,190,191], and nervous system tumors [192]. Radiosensitive tumors exhibit less benefit by combining TRAIL agents [184]. Similarly, combining a TRAIL-CD19 fusion protein with radiation showed efficacy in acute lymphocytic leukemia [193]. To date, these studies are limited to pre-clinical.

## 4. Intrinsic Pathway

### 4.1. Pro- and Anti-Apoptotic Regulators within the Pathway

The cell-intrinsic pathway is also known as the ‘mitochondrial pathway’ since this involves the changes in the mitochondrial membranes and release of proteins that result in widespread protein proteolysis and DNA cleavage. BH3-containing pro-apoptotic proteins operate as antagonists of the anti-apoptotic proteins [194]. In the intrinsic pathway, when cells are stressed by intrinsic lethal stimuli, BID, BAX, BAK, and lipids cooperate to form supramolecular openings in the outer mitochondrial membrane, which is called mitochondrial outer membrane permeabilization (MOMP) [7]. In the intrinsic pathway, MOMP is regulated by interactions of Bcl2 family proteins, which play diverse anti- and pro-apoptotic roles. BCL2 family proteins are categorized into the following groups: Multi-domain pro-apoptotic proteins BAX and BAK (BAK1); BH3-only proteins BIM (BCL2L11), BID, and PUMA (BBC3); BH3-containing pro-apoptotic proteins, BAD, HRK, BCL-XS (a short splice variant of BCL2L1), BIK; anti-apoptotic proteins BCL2, BCL-XL (a long splice variant of BCL2L1), MCL1, BCLW (BCL2L2), BFL1 (BCL2A1), and BCL-B (BCL2L10). BCL-2 or BCL-XL overexpression, loss of BAX or BAK contribute to the resistance of cells to the apoptosis [195]. While multi-domain pro-apoptotic proteins, such as BAX and BAK, directly promote MOMP, BH3-only proteins support MOMP via agonistic effects of pro-apoptotic proteins and antagonistic effects of anti-apoptotic proteins.

Following MOMP, mitochondrial intermembrane proteins, such as cytochrome c and SMAC/direct IAP-binding protein with low PI (DIABLO), are released into the cytosol [196], promoting the assembly of a caspase-activating complex apoptosome, which also includes apoptotic peptidase activating factor 1 (APAF1), Deoxyadenosine triphosphate (dATP), and pro-caspase 9. The inhibitor of the apoptosis domain known as the IAP repeat and baculovirus inhibitor of apoptosis protein repeat (BIR) is a critical motif of IAP proteins. Upon release of mitochondrial SMAC/DIABLO into the cytosol, IAP proteins, such as XIAP, cIAP1, cIAP2, and survivin, that directly inhibit caspase 3, 7, and 9 are inactivated, leading to apoptosis of cells [197]. cIAP1 and cIAP2 also function as positive regulators of the NF-κB pathway. SMAC mimetics cause degradation of the cIAPs and this contributes to apoptosis. In fact, it is not clear how much direct caspase inhibition is a target of these two. XIAP does inhibit caspases and is in turn inhibited by IAP inhibitors. Increased IAP expression causes the resistance to apoptosis in a variety of cancers. Overexpression of XIAP, cIAP1, cIAP2, and survivin have been reported in the chemotherapy resistant cancer cells, and the inhibition of IAP members sensitize cells to apoptosis induced by cytotoxic agents [198]. IAP proteins contain a c-terminal RING domain with E3 (ubiquitin ligase) activity. However, survivin lacks a RING domain, but it interacts with multiple proteins, e.g., Rb, p53, IGF-1, and EGF proteins [199,200]. 

### 4.2. B Cell Lymphoma-2 (BCL-2)

#### BCL-2 Inhibitors 

BCL-2 inhibitors have been approved for their use as an anti-cancer drug by the U.S. Food and Drug Administration (FDA) as the first agent among all intrinsic apoptosis inhibitors (Table 2). Hematological malignancies were the first to receive this clinical indication. For example, BCL-2 over-expression contributes to resistance in chronic lymphocytic leukemia (CLL) significantly. While translocation of t(14:18) that causes transcriptional upregulation of BCL2 is a relatively a rare event in CLL, BCL-2 is often over-expressed by various mechanisms in CLL.

Venetoclax (ABT-199) is the first class-agent approved by the FDA. Venetoclax inhibits BCL-2 by binding to the protein, displacing pro-apoptotic heterodimer partners, e.g., BIM. It binds to BCL2 protein with a higher affinity, compared to other BCL2 family members, such as BCL-w or BCL-XL. The first approved indication was in therapy resistant CLL, where venetoclax induced a 79% response rate as a single agent [201]. The response remained as high as 77% in the CLL with 17p deletion, leading to an accelerated FDA approval of venetoclax in 17p deletion CLL [201]. The overall response rate (ORR) of venetoclax and rituximab combination was higher at 86%, providing a combination strategy [202].

Venetoclax also showed efficacy in acute myeloid leukemia (AML). In elderly patients (median age 74) with treatment naïve AML who are ineligible for intensive chemotherapy, the combination of venetoclax and cytarabine showed an impressive 61% ORR, including 21% complete responses (CRs). In this study, patients with *TP53* mutation achieved 44% CR, higher than other therapeutics tested [208]. Patients with a good prognostic marker *NPM1* mutation had 100% CR. Given this superb efficacy, the FDA granted a break-through designation for venetoclax allowing expedited future study in chemotherapy-ineligible AML. With this notable efficacy demonstrated in hematological malignancies, venetoclax is now being studied in solid cancers. In breast cancer, venetoclax is combined with faslodex to treat endocrine therapy resistant hormone receptor positive breast cancer, [209] given that functional estrogen receptor (ER) transcriptionally up-regulates BCL-2. *Navitoclax (ABT-263)* inhibits both BCL-w and BCL-XL, in addition to BCL-2. A phase II clinical trial to test navitoclax in small cell lung cancer (SCLC) so far has shown 1 out of 39 patients with a partial response (2.6%), along with 1.5 months median PFS and 3.2 months OS [210]. A combination with cytotoxic chemotherapy resulted in poor tolerability due mostly to thrombocytopenia. Therefore, the drug development strategy using the chemotherapy combination with ABT263 was terminated (no reference given due to negative clinical results). Instead, ABT263 in combination with a targeted agent seems to be well tolerated. Ongoing studies include the combination of navitoclax and vistusertib in SCLC*, osimertib in EGFR mutated NSCLC*, sorafenib in therapy refractory solid tumors in phase I [211], MEK inhibitor trametinib combination in phase Ib for all solid tumors*, and with trametinib and dabrafenib in BRAF mutated melanoma [212]. The results are pending (current ongoing trials summarized in Table 2).

Obatoclax mesylate (known as GX15-070) is another BCL-2 inhibitor, and the combination of obatoclax with radiation showed tumor control in nasopharyngeal carcinoma [213], prostate cancer [214], and leukemia [215]. Unfortunately, clinical development was discontinued given high-profile toxicities, such as ataxia and thrombocytopenia [216,217]. 

Gossypol (AT-101) is another BCL-2 inhibitor, a natural phenol derived from a plant, that has been used as a contraceptive and anti-malarial drug. It inhibits multiple dehydrogenase enzymes leading to G0/G1 arrest, as well as BCL-2 family members. It has shown synergistic effects with radiation in head and neck cancer [213] and with an AR inhibitor in prostate cancer [218]. However, it also is noted to induce autophagy related cancer survival when given as a single agent [219,220]. It has been studied in chemotherapy resistant breast cancer with disappointing clinical activity [221], as well as in glioblastoma [222]. Thus, the clinical development has been discontinued at the present time.

S55746/BCL201 is a new generation BCL-2 inhibitor, and has shown efficacy in therapy refractory CLL, with favorable pharmacokinetics [223]. Further clinical trial results are pending. 

### 4.3. Challenges and Strategies to Improve the Efficacy of BCL-2 Targeted Therapy

#### 4.3.1. Toxicity of BCL-2 Inhibitors

Thrombocytopenia is a known toxicity of this class agent. In a pre-clinical study, venetoclax was found to be less toxic to the survival of platelets [224]. However, in phase I/II trials of venetoclax, dose limiting toxicity (DLT) was still a grade 4 thrombocytopenia. Nausea, anemia, febrile neutropenia, and fatigue were also observed in greater than 30% of patients, but mostly as grade 1/2 [208]. Both BCL-XL and BCL-2 contribute to the survival of megakaryocytes (along with MCL1) [225]. Exposure of platelets to navitoclax, which inhibits both BCL-2 and BCL-XL, depleted intracellular calcium deposit within the cells, leading to impaired platelet function as well [226]. In addition, unexpected neurological toxicities were observed in pan-BCL-2 inhibitors. 

#### 4.3.2. Resistant Mechanisms

Pro-apoptotic proteins, such as BIM, NOXA, and PUMA, form heterodimer complexes with BCL-2 family members to inactivate them. However, when the heterodimers get dissociated, the other anti-apoptotic proteins can capture the released pro-apoptotic proteins, thus resulting in inhibited apoptosis. This can be within the BCL-2 family members or other BH3 domain containing proteins. For instance, BCL-XL specific inhibitors, A-1155463 and A-1331852, have been studied in preclinical models and did not induce apoptosome formation, and yet have shown synergy with BCL-2 inhibitors [227]. Lastly, BCL-2 inhibitors induced release of Beclin-1 to promote an autophagy [228].

#### 4.3.3. Combination Strategies

Co-inhibition of compensatory pro-apoptotic molecules is one key to combination strategies. For example, selective BAX trigger site activator (BTSA)1 [229,230] can be combined with a BCL-2 inhibitor to induce the activation of pro-apoptotic protein BAX. In addition to several combination partners that co-inhibit compensatory apoptosis, regulators have been suggested to exhibit synergy. Venetoclax and Bromodomain and extra-terminal motif (BET) inhibitor ABBV-075 in SCLC cell lines showed synergy by direct up-regulation of BIM by BET inhibition [231].

RAS-MAPK pathway inhibition causes the up-regulation of anti-apoptotic proteins, supporting this as a potential co-inhibition strategy [232]. As mentioned earlier, there are multiple trials to test targeted agents with navitoclax. A phase I/II trial of the navitoclax + trametinib combination (phase II with various *KRAS* mutated tumors) was initiated based on the preclinical data showing that ABT263 binding to BCL-XL, which inhibits pro-apoptotic proteins, in combination with a MEK inhibitor led to dramatic apoptosis in many KRAS driven tumors [233]. Trials of other MEK inhibitor and BCL-2 inhibitor combinations—navitoclax + sorafenib [211] and navitoclax + trametinib + dabrafenib [212]—are also ongoing. If successful, this combination strategy may be applied to other agents within the class [232].

### 4.4. Inhibitor of Apoptosis Proteins (IAPs)

IAP family inhibitors augment apoptosis via induction of proteasome degradation of cIAP1 and 2. cIAP1 and 2 are RING finger E3s, which have three N-terminal BIR domains and a c-terminal RING finger domain. IAP antagonists bind to the BIR domain and trigger rapid RING finger dependent auto-ubiquitination, thus the RING finger domain is critical for cIAP 1 and 2. Upon binding of IAPs to the TNF receptor, the IAPs shift signaling of the TNF receptor to the canonical anti-apoptotic NF-κB pathway rather than the activation of caspase-8, thereby blocking apoptosis [234,235,236,237,238,239]. Thus, inhibition of IAPs along with the TNF-α mediated activation of the TNF receptor could result in synergistic activation of the extrinsic apoptosis pathway [240]. XIAP is a family member of the IAP proteins, but is the only bona fide direct caspase inhibitor by uniquely carrying the IAP-binding motif (IBM)-interacting groove and an inhibitory element unlike other family members [241]. XIAP directly inhibits caspase 3 and 9 activation, thus antagonizing the intrinsic and extrinsic apoptosis pathways. The tetrapeptide binding motif of the SMAC N-terminus and the groove on the IAP protein surface are involved in the interaction of these SMAC and IAP proteins [242]. Unlike IAP1, IAP2, and XIAP, survivin lacks the c-terminal RING finger domain, but instead interacts with various proteins that regulate the transcription activity of genes.

### 4.5. IAP Inhibitors 

LCL161 is a SMAC mimetic IAP inhibitor, studied in plasma cell myeloma [243] and ALL [244], and shows pre-clinical activity as both a single agent and in combination with other therapies. In a small phase II trial of primary myelofibrosis, it showed 38% ORR in a previously treated patient who failed more than two lines of therapy, and was shown to be safe [245]. It is now being studied in other early-phase clinical trials [246,247,248]. Currently studied IAP inhibitors are summarized in Appendix A.

Birinapant (TL32711) is a small peptido-mimetic of SMAC that causes degradation of cIAP1, inducing caspase-mediated apoptosis [166,249]. The drug causes down regulation of IAP1 and this switches the TNF pathway from. *NF-κ* to activation of caspase 8 and extrinsic apoptosis [249]. The combination of demethylating agents and birinapant showed synergy in leukemia cells [250]. It also induced radiation sensitization activity in colorectal cancers [251] and head and neck cancers [249] in pre-clinical models. Currently, it is being studied in ovarian cancer, head and neck, leukemia, and lymphoma as single or combination therapy [252,253].

ASTX660 is a novel dual cIAP/XIAP antagonist that binds to the BIR3 domain of XIAP, and thus has dual antagonistic activity to both cIAP and XIAP. This molecule successfully interrupted the binding of caspase 9 and SMAC to the XIAP, and has been shown to potentiate TNF-α-mediated apoptosis in melanoma and breast cancer cell lines [254]. It is currently in early development in a phase I study of solid tumors and lymphomas [255].

CUDC-427 (GDC-0917) is a selective oral antagonist of IAP and has displayed exciting clinical activity. A phase I trial demonstrated an encouraging response, including a patient with a mucosa-associated lymphoid tissue (MALT) lymphoma and a patient with platinum-refractory ovarian cancer who both attained a CR [256].

XIAP-AS (AEG35156) is an antisense oligonucleotide against XIAP that has shown pre-clinical activity in leukemia [257]. However, a clinical trial in leukemia revealed disappointing results [258], perhaps because of poor tissue penetration and rapid degradation.

Embelin is a natural plant-derived agent that has been identified as an inhibitor of XIAP, but did not have favorable bioavailability, lacking efficacy as a single agent [259].

Sepantronium bromide (YM-155) is a small molecule inhibitor of survivin, by direct inhibition of the protein and transcriptional suppression of the gene, *SOCG_03840*. It showed efficacy against lymphoma when combined with the B cell inhibitor antibody, rituximab [260].

LY2181308 is a survivin-antisense oligonucleotide and showed pre-clinical activity, when electroporated into leukemic target cells, but a phase I and II clinical trial showed no significant activity [261,262].

SurVaxM (SVN53-67/M57-KLH, DRU-2017-5947) is a 14 amino acid peptide from the human survivin protein sequence and serves as a neo-immunogen that initiates an immune response to survivin containing tumor cells, and thus can sensitize cancer cells to immune mediated apoptosis by targeting survivin that is differentially over-expressed in tumor cells compared to normal cells [263,264,265]. It was studied in patients with glioma as a single agent, and in combination with temozolomide in glioblastoma, showing impressive OS [263]. This resulted in accelerated approval to be further studied by the FDA.

*DPX-Survivac*, and *Sur1M2* are also in clinical development to target survivin (Table 2).

### 4.6. Challenges and Strategies to Improve the Efficacy of IAP Targeted Therapy

#### 4.6.1. Toxicity of IAP Inhibitors

Observed class-agent on target side effects include cytokine release syndrome (CRS), due to their ability to induce TNF-α mediated immune reactions [246], confirmed to be the dose limiting toxicity (DLT) in the clinical setting. However, in the recommended phase II dose, these agents were mostly well tolerated. The DLT of birinapant is headache, nausea, and vomiting, as a mild form of CRS [266]. Peripheral neurological disorders, such as Bell’s palsy, have also been reported [266].

#### 4.6.2. Resistant Mechanisms

Compensatory up-regulation of other IAPs, survivin, and XIAP occur when c-IAP1 and 2 are inhibited, serving as a key resistance mechanism as well, suggesting the need of a co-inhibitory strategy.

#### 4.6.3. Combination Strategy

In the use of IAP inhibitors as an apoptosis induction strategy, the co-inhibition of X-linked inhibitor of apoptosis protein (XIAP) and survivin, which can interact with various cascades, should also be considered as a potential synergistic combination [267]. While the occurrence of CRS as a class-agent side effect deserves clinical attention, the ability to induce an immune reaction can be a rationale to combine IAP inhibitors with check point inhibitors as another potential synergy partner. Indeed, the combination of the SMAC mimetic LCL161 along with two check point inhibitors, anti-TIM3 and anti-PD1, in a pre-clinical study (triple combination) showed efficacy of LCL161 in syngeneic murine colon cancer [268].

### 4.7. Myeloid Leukemia Cell Differentiation Protein 1 (MCL1)

MCL1, a multi-domain BCL2 family pro-survival anti-apoptotic protein, is difficult to develop as an inhibitor given a key binding pocket is hydrophobic. However, several agents have been successfully developed, and the first agent is being tested in the clinic. MCL1 dimerizes with pro-apoptotic proteins, such as BAK or BAX. on the mitochondrial membrane and sequesters pro-apoptotic proteins in a similar fashion as other pro-survival molecules, such as BCL-2 or BCL-XL, to prevent MOMP [269]. Overexpression of MCL1 is implicated as a resistance factor for chemotherapeutics, such as paclitaxel and vincristine, by potentially restoring anti-apoptotic signaling and decreasing the sensitivity to chemotherapy in breast cancers [270,271,272]. Thus, it is an important and now druggable target. MCL1 has shown significant clinical relevance in triple-negative breast cancer (TNBC), being commonly amplified in 56% of TNBC tumors, and with overexpression associated with a high tumor grade and poor prognosis [273,274]. Furthermore, MCL1 was noted to be frequently amplified in endocrine therapy resistant breast cancer, and prostate cancer [275,276]. However, the recent advancement of medicinal chemistry led to the development of direct MCL1 inhibitors. Another means to inhibit MCL1 is by inhibition of CDK9. CDK9 is a cyclin dependent kinase which transcriptionally regulates key genes, such as MYC, MCL1, and cyclin D. While CDK9 inhibition results in the decreased transcription of many genes, MCL1 protein can be more dramatically affected due to the short half-life of the protein, which results in a higher dependency of protein levels on gene transcription. Thus, inhibitors of CDK9 have been studied as one of the key indirect MCL1 targeted therapeutics.

### 4.8. MCL1 Inhibitors 

AMG 176 is a first-in-class MCL1 inhibitor, binds with high affinity and selectivity to the BH3-binding groove of MCL1, and has shown efficacy in pre-clinical studies [277]. Recently, the synergy between AMG176 and the BCL2 inhibitor, venetoclax, was confirmed in a pre-clinical AML model [278]. Based on these pre-clinical findings, this molecule is currently undergoing testing in a phase I trial in multiple myeloma [279] and AML [280].

AZD5991 is another specific inhibitor of MCL1 that induces apoptosis at low nanomolar concentrations in MCL1-dependent multiple myeloma cell lines. However, it can lose its activity upon overexpression of BCL-XL or small interfering RNA (siRNA)-mediated knockout of BAK [281]. It is under early clinical development in a phase I study.

S63845 has a high affinity to the BH3-binding groove of MCL1, showed preclinical activity against multiple myeloma, leukemia, and lymphoma cells [282], and is currently being tested in phase I studies.

CDK9 inhibitors: A number of trials are currently testing CDK9 inhibitors in both hematological and solid tumor, as summarized in the Appendix A (since no results are available).

### 4.9. Challenges and Strategies to Improve the Efficacy of MCL1 Targeted Therapy

#### 4.9.1. Toxicity of MCL1 Inhibitors

Given the early stage of development of MCL1 inhibitors, comprehensive toxicity/safety profile of these agents is not readily available to date. However, some of the reported toxicities include cytopenia, diarrhea, and fatigue. More toxicities are to be expected as the early phase clinical trials mature.

#### 4.9.2. Resistant Mechanisms

Resembling that of BCL-2 inhibitors, MCL1 inhibition can have cross-compensatory activation of other anti-apoptotic molecules, which can replace MCL1 to suppress pro-apoptotic molecules. Additionally, MCL1 is known to cross-talk with multiple signal transduction pathways, e.g., EGFR and PI3K-AKT, thus cancer cells can utilize other survival mechanisms when MCL1 is suppressed, as possible resistance mechanisms. These mechanisms need to be further investigated.

#### 4.9.3. Combination Strategy

Several partners can be rationalized as combination therapeutics with an MCL1 inhibitor. Down-regulation of MCL1 has been noted to be implicated in preclinical and phase I clinical trials as the primary mechanism of activity of alvocidib (flavoporidol), a pan-CDK inhibitor, in patients with CLL or AML [283,284]. This molecule inhibits many, including CDK1,2,4,6,7,9. The second category of agents that can have synergy with the MCL1 inhibitor in therapy resistant breast cancer are other BCL2 family (BCL2, BCL-XL, BCL-w) inhibitors. In hematologic malignancies, BCL-2 is often found to be over-expressed or amplified, causing evasion of apoptotic cell death. In solid cancers, IAP family and MCL1 have been noted as more key contributors of resistance to conventional therapeutics. MCL1 has a high-affinity protein-protein interaction with Bcl-2-like protein 11 (BIM), and it is hard to create a molecule that can directly interfere with this strong interaction. [285]. HDAC inhibitors induce the mRNA level of NOXA and PUMA, thus a strategy to use the HDAC inhibitor to suppress MCL1 has been developed, before direct MCL1 inhibitors were available [286]. Co-inhibition of MCL1 and BCL-2 could be an effective therapeutic approach, however, there is concern for a potential increased toxicity. Thus, a careful approach is required in studying such a combination, as shown in pre-clinical data [287].

More recently, the biologic connection between mucin 1 (MUC1) and anti-apoptotic proteins has been reported. MUC1 is a glycoprotein that has shown to be highly expressed in aggressive cancers and noted to be over-expressed in TNBC even in early stages [288]. It has been shown to play a critical role in glutamine metabolism [289], and epigenetic modulation via RNA polymerase II [290]. MUC1, a single chain peptide that undergoes endoplasmic reticulum mediated cleavage into the extracellular vs. cytoplasmic domain, utilizes tight regulation as to how it can be divided into two units. Interestingly, the C terminal unit of MUC1 was shown to be up-regulated in TNBC cells that were resistant to BCL-2 inhibitors but sensitive to an MCL1 inhibitor, suggesting that the C terminal domain of MUC1 can mediate resistance to BCL2 inhibitor induced apoptosis. This suggests that there could be synergy between BCL-2 and MCL1 inhibitors. Inhibition of this specific MUC1 subunit either by siRNA or an inhibitor in a pre-clinical model resulted in the suppression of phosphorylation mediated activation of ERK and AKT [283,284], resulting in effective apoptosis [291]. In addition, this biological mechanism of MUC1 and MCL1 regulation supports the exploration of combination therapy using anti-metabolic therapeutics (e.g., OxPhos inhibitor or glutamate inhibitor).

## 5. Development of Biomarkers of Apoptosis

### 5.1. Biomarkers to Predict TRAIL Sensitivity

Preclinical studies have demonstrated that only a subset of cancer cells is sensitive to TRAIL, while most tumors are TRAIL-resistant [292,293,294]. Similarly, the case reports from clinical trials showed the benefit of DR targeting therapy in some patients [295,296]. In past clinical trials with DR agonist therapies, no patients were selected based on predictive biomarkers. The utilization of patient selection criteria will help to identify who may benefit from DR agonist therapy. Ongoing efforts to identify this target group suggest that TRAIL sensitivity biomarkers are context-dependent. While a lack of TRAIL receptors predicts resistance to TRAIL targeted therapies as shown in neuroblastoma [297], there is no definite correlative link between total DR protein expression levels and the sensitivity of TRAIL [51,163,298]. In addition, mRNA levels of DR4/5 do not reflect their functional protein expression due to the complex post-translational modifications, such as glycosylation and trafficking of the synthesized receptors, [299]. Here, we list the recent findings relevant to potential TRAIL sensitivity biomarkers.

O-glycosylation of DR5 is required for full activity of DR5, and the expression level of O-glycosylation (O-glyc) genes is associated with TRAIL sensitivity [300]. However, O-glyc genes were not overexpressed in TRAIL-sensitive triple negative breast cancer (TNBC) cells [44,51], indicating that O-glyc may be a TRAIL sensitivity biomarker in specific cancer cell types.

N-glycosylation of DR4 plays an important regulatory role for DR4-mediated apoptosis [301,302]. Cells expressing N-glycosylation (N-glyc)-defective mutants of DR4 were less sensitive to TRAIL than their wild-type counterparts. Defective apoptotic signaling by N-glyc-deficient DR4 was associated with lower DR4 aggregation and reduced DISC formation, but not with reduced TRAIL-binding affinity.

Decoy receptors DcR1/2 co-expressed with DR on the same cell can block the transmission of the apoptotic signal block TRAIL sensitivity [154,303]. TRAIL binds decoy receptors *(DcR1/2)* but does not induce apoptosis since these decoy receptors either lack (DcR1) or have a truncated (DcR2) cytoplasmic death domain [44]. While these receptors can inhibit TRAIL-mediated apoptosis by competing with the active receptors for ligands, their role in the TRAIL resistance of tumor cells is unclear [44].

c-FLIP is a major anti-apoptotic regulator that inhibits cell death mediated by DR [304,305,306]. Three splice variants of c-FLIP function at the DISC level by blocking the processing and activation of procaspase-8 and -10. Overexpression of c-FLIP has been identified in many different tumor types, and its downregulation in vitro has been shown to restore TRAIL-mediated apoptosis [307,308,309]. 

Mesenchymal or epithelial subtype of cancer: Mesenchymal TNBC cell lines are extremely sensitive to TRAIL-induced apoptosis compared with other subtypes of breast cancer cell lines [44,51]. In contrast, pancreatic, lung, and colon cancers cell lines with an epithelial phenotype are more TRAIL sensitive than those with a mesenchymal phenotype [310]. The mechanistic basis of these findings is yet to be established. 

Caspase-8 expression level [311] or activity [312] is associated with TRAIL sensitivity in some cancer cell types. 

Gene signature: Using genome-wide mRNA expression profiles, a study has identified 71 genes whose expression levels are systematically higher in TRAIL-sensitive cancer cell lines [313]. This signature accurately predicts TRAIL sensitivity in a total of 95 human cancer cell lines [313]. The gene signature is dominated by IFN-induced genes and the MHC genes [313].

Cell metabolism: Several reports indicate that manipulating cell metabolism, such as glutamine deprivation [314], methionine depletion [315], and metformin [316], can sensitize TNBC cells to DR agonists. A recent study suggest that mitochondrial content is a useful biomarker for the prediction of apoptotic susceptibility [317].

At this point, no predictive biomarkers have been successfully applied in clinical trials.

### 5.2. Biomarkers to Predict Sensitivity to Intrinsic Apoptosis-Targeted Agents

The development of robust predictive biomarkers allows proper validation of target-based efficacy as well as the appropriate selection of the target patient population [318].

#### BH3 Profiling

One recent key strategy of developing a biomarker utilizes the BH3 domain, which is included as a diverse member of the BCL-2 family [319]. Given the importance of the BH3 domain containing molecules within the network cascades in apoptosis, the finger printing using a synthetic BH3 peptide (BH3 profiling) was developed first in AML [320]. Letai et al. developed a BH3 profiling strategy to detect the fingerprints of each tumor and their dependency on different members of the BCL-2 family, via ex vivo testing as a measure of apoptotic sensitivity. This BH3 profiling uses a functional single cell-based analysis to capture the mitochondrial apoptotic sensitivity as priming, using synthetic BCL-2 BH3 domain-like peptides [321]. A synthetic peptide that mimics the BH3 domain of BCL-2 protein family members is added to a single cell with access to the mitochondria to induce the cascade mediated apoptosis. Then, the occurrence of MOMP is measured by loss of mitochondrial potential, using tetramethyl rhodamine or immunofluorescence-based measurements [322,323]. The identity of the BH3 peptide that induces MOMP indicates which BCL2 family member the cell is dependent upon and could help identify the best agent for use in that tumor. Given the complexity of the intra-/inter networking of these apoptosis regulator proteins, comprehensive interrogation of the pathway as a whole could provide better prediction to apoptosis regulator targeted therapeutics. For example, the combination of the BCL-2 inhibitor and MCL1 inhibitor was synergistic when cancer cells were either dependent on MCL1, or the BCL-2 family contributed to the emerging resistance. A specific test to predict such resistance dependency can thus aid the selection of a patient population who benefits from each therapeutic.

Pro- and Anti-Apoptotic Regulator Heterodimer Detection measures the status of BIM or MCL1 trapping by the anti-apoptotic family and can be reversed by the BCL-2 inhibitor, and can predict the sensitivity to a specific inhibitor. This functional measurement of the apoptotic state is critical because an accurate measurement of available pro-apoptotic Bim and the proportion of the heterodimer of Bim and other partners can measure the actual scope of apoptosis better than a simple measurement of IHC based apoptotic regulators, when multi-domain anti-apoptotic Bcl-2 family (candidate epitopes that can bind to Mcl-1 and BH3) only peptide forms a heterodimer complex, by using counter-screen (Figure 2).

### 5.3. Biomarkers for Pharmacodynamic and Downstream Effect 

Caspase-3 is an indirect reflection of apoptotic activities and has been most commonly used for both measuring ‘cell death’ and ‘apoptosis’ as an intermixed result endpoint. In colorectal cancer, the induction of caspase 3 occurs when beta-catenin and COX2 are activated, suggesting that baseline activation of caspase may be a surrogate marker of activated compensatory pathways [324], thus not precisely measuring the intrinsic or extrinsic apoptosis pathways’ activation. This limitation can encounter more challenges when the assay is used for testing direct apoptosis inducing agents to measure an on-target-effect. One group has developed a signature of activated caspase 3 inducing changes and used this as an imaging probe for a positron emission tomography (PET) scan [325]. Authors measured the insoluble CD44, cytoplasmic HMGB1, and several other proteins that are activated when caspase 3 is activated, as a surrogate marker of caspase 3 activity. This novel approach could be expanded in further applications in imaging detection of apoptotic activity.

#### ELISA Based Protein Assay

Recently, the US National Cancer Institute developed an apoptosis multiplex immunoassay panel designed to detect the pharmacodynamic change among the constituents of the apoptosis pathways. Several proteomic-study-based panels are under development for detecting the dynamic change of apoptosis-related molecules; however, current assays require flash frozen tissue that can be processed immediately, in a large quantity, which possibly limits the wider use of this method to predict an apoptotic agent [326]. The M65 ELISA measures levels of human full-length and caspase-cleaved CK18 (cCK18) in the supernatant or serum, measuring overall cell death. The M30 ELISA measures the level of cCK18, identifying the fraction due to activation of apoptosis [149]. The differential staining cytotoxicity (DiSC) assay is one of the assays developed to measure the sensitivity/resistance to therapeutics by detecting the cell killing activity [327]. Only a few cells are required to be tested, as long as the cancer and non-cancer cells can be separated in the cell suspension. The technical difficulties associated with performing this assay makes it challenging for it to be used widely. Lastly, an assay to detect the cIAP degradation from both tissue and blood using a capillary electrophoresis based mini-Western system has been developed as a PD biomarker of on target SMAC mimetic activity [252] (Cancer Therapy Evaluation Program Biomarker Review Committee (CTEP BRC) approved).

## 6. Conclusions and Future Directions

Apoptosis is a complex process that involves a cascade of molecules that interact via similar binding sites existing in different status, which makes not only the development of inhibitors but also the detection of ‘apoptotic homeostasis status’ challenging. In addition, pro- and anti-apoptotic molecules interact with several other signal transduction pathways, including PI3K-Akt and MAPK, and induce autophagy to further complicate the detection of changes in downstream events [328]. As more direct apoptosis-targeted therapeutics are tested in the clinic, investigators will need to develop robust biomarkers to predict the sensitivity of cancer to either a single agent or combination. In addition to the well-designed effective inhibitors to target various nodes and molecules within the apoptosis cascade, the companion diagnostic tool development is also a key to the success of these agents. Biomarker assays are critical for the translation of apoptosis-targeted agents as they enable both selection of an appropriate target population, as well as monitoring of the effectiveness of new therapeutic agents that have been developed to target this pathway.

To overcome these challenges, we suggest some approaches to make apoptosis-targeting successful. First, we need to develop synergistic synthetic lethal combinations of direct apoptosis-targeted agents and agents that co-inhibit a compensatory pathway. Well thought-out pre-clinical studies to unravel the precise target of each agent, in addition to an elucidation of emerging resistance mechanisms within a complex apoptosis network, is critical in the development of synergistic partners. Third, in depth medicinal chemistry support to elucidate the binding structure and targeting affinity of direct apoptotic regulator inhibitors/agonists while achieving a wide therapeutic window to minimize the toxicity. The recent success of BET domain inhibitors, a class agent that previously was considered as a non-druggable target, highlights the importance of this field to the development of apoptosis inducing agents. Fourth, we propose the need to develop a robust panel of assays that can identify a predictive biomarker of response to either single or combination therapy using BH3 profiling or other novel approaches to detect heterodimers that can measure the status/balance among various heterodimers (Figure 2). Second, we need better strategies to co-inhibit compensatory co-regulators within each apoptosis induction pathway (Figure 3). Lastly, patient derived sample collection and analysis in the context of novel anti-apoptosis therapy in the clinical setting will provide a direct evidence from the bedside back to the bench development strategy (Figure 4). All of these strategies require diversified efforts encompassing all areas of drug development as a community.

It is notable that the dependency of each cancer to apoptosis pathway regulation is heterogeneous, underscoring that single agent activity will likely be limited. Toxicities and lack of anti-cancer activity of older generation apoptosis-targeted therapeutics has given a negative image to the apoptosis-targeted agent as an effective anti-cancer therapeutics approach. However, thanks to the further development of medicinal chemistry, bioinformatics, biostatistics to support continued therapeutic, and biomarker discoveries, researchers are closer to the translation of apoptosis-targeted agents than ever before. The success story of BCL-2 inhibitors in hematologic malignancies is quite encouraging, along with new MCL1 and IAP inhibitors that are actively being tested in the clinic. We hope this timely review can further help the development of apoptosis-targeted therapeutics as a novel approach that can overcome resistance to conventional therapeutics in cancer.

## Figures and Tables

**Figure 1 cancers-11-01087-f001:**
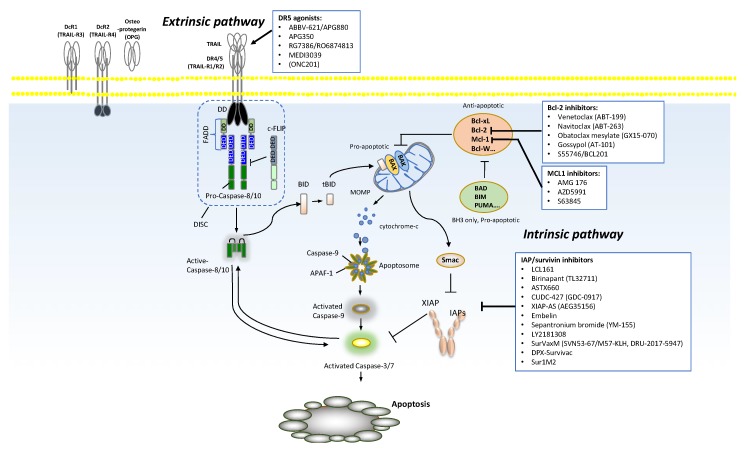
Two major apoptotic pathways.

**Figure 2 cancers-11-01087-f002:**
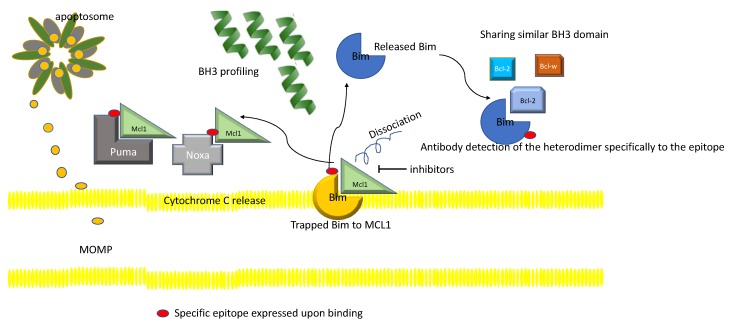
Interrelationship among BH3 domain containing proteins, including MOMP and predictive biomarkers.

**Figure 3 cancers-11-01087-f003:**
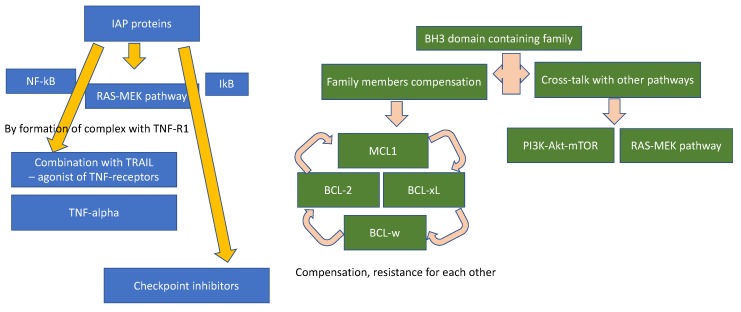
Combination strategy based on the reported pathway that interacts with apoptosis regulating proteins.

**Figure 4 cancers-11-01087-f004:**
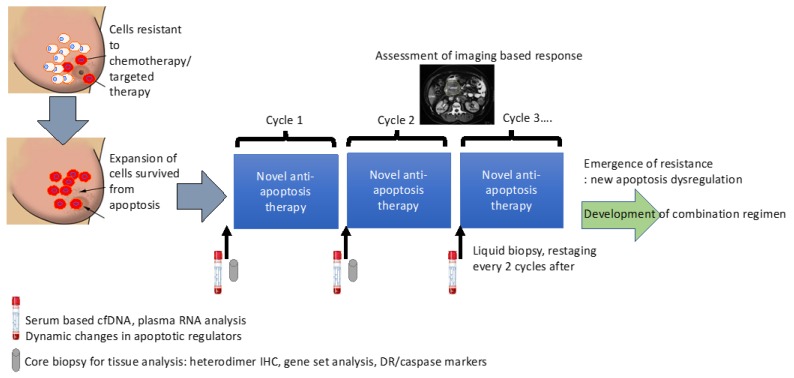
Proposed strategy to translate the anti-apoptosis therapeutics.

**Table 1 cancers-11-01087-t001:** Current ongoing clinical trials testing second and third generation death receptor targeted therapeutics.

Agents	Molecule Type and Target	Clinical Trial Phase	Current Status	Combinatorial Agents	Target Disease	Clinical Trial Number
ABBV-621/	fusion protein monomer	I	recruiting	venetoclax (DLBCL, AML only)	previously treated solid tumors and hematologic malignancies	NCT03082209
APG880	DR 4/5					
APG350	fusion protein monomer	Preclinical yet		Single agent	solid tumors (colon, pancreatic cancer) in mouse xenograft model	
	DR 4/5					
RG7386/	bispecific antibody	I	completed	Single agent	locally advanced or metastatic solid tumors	NCT02558140
RO6874813	DR5					
TAS266	tetravalent nanobody	I	terminated	Single agent	solid tumors	NCT01529307
	DR5					
MEDI3039	multivalent scaffold protein superagonist	Preclinical yet		Single agent	solid tumors (breast, colon) in mouse xenograft model	
	DR5					
HexaBody^®^-DR5/DR5 (GEN1029)	DR5	I/II	recruiting	Single agent	solid tumors	NCT03576131
CPT	circularly permuted TRAIL	Ia	completed	Single agent	lung cancer, colon cancer, lymphoma, multiple myeloma, etc.	ChiCTR-ONRC-12002084 (http://www.chictr.org.cn/)
		Ib	completed	Single agent	relapsed or refractory multiple myeloma	ChiCTR-TNRC-12001896
		II	completed	Single agent	relapsed or refractory multiple myeloma	ChiCTR-ONC-12002065
		IIa	completed	Single agent	multiple malignant solid tumors	ChiCTR-ONRC-12002086
		II	completed	+ thalidomide	relapsed or refractory multiple myeloma	ChiCTR-ONC-12002066
		II	recruiting	CPT + thalidomide and dexamethasone (TD) or TD alone	relapsed or refractory multiple myeloma	ChiCTR-TRC-11001625
		III	recruiting	CPT +/− TD	relapsed or refractory multiple myeloma	ChiCTR-IPR-15006024
ONC201	small molecule compound	I	recruiting	Single agent	recurrent H3 K27M-mutant glioma	NCT03416530
		I	recruiting	Single agent	solid tumors	NCT02324621
		I	recruiting	Single agent	solid tumors, multiple myeloma	NCT02609230
		I	suspended	Single agent	solid tumors	NCT02250781
		I/II	recruiting	Single agent	multiple myeloma	NCT02863991
		I/II	recruiting	ixazomib, dexamethasone	multiple myeloma	NCT03492138
		I/II	recruiting	+/− cytarabine	relapsed/refractory acute leukemia and high-risk myelodysplastic syndromes	NCT02392572
		I/II	recruiting	Single agent	relapsed/refractory non-Hodgkin’s lymphoma (NHL)	NCT02420795
		I/II	withdrawn	N/A	glioblastoma	NCT02038699
		II	recruiting	Single agent	recurrent/refractory metastatic breast cancer and advanced endometrial carcinoma	NCT03394027
		II	recruiting	Single agent	recurrent or metastatic endometrial cancer	NCT03099499
		II	recruiting	Single agent	recurrent or metastatic type II endometrial cancer endometrial cancer	NCT03485729
		II	recruiting	Single agent	neuroendocrine tumors	NCT03034200
		II	recruiting	Single agent	recurrent glioblastoma and H3 K27M-mutant glioma	NCT02525692
		II	recruiting	Single agent	recurrent H3 K27M-mutant glioma	NCT03295396

**Table 2 cancers-11-01087-t002:** Current ongoing clinical trials testing BCL2 inhibitors, ABT-199 (venetoclax), and ABT-263 (navitoclax), as a single agent or in combination.

Agents and Ongoing Trials	Target Disease	Phase of the Trial	Combinatorial Agents	Supportive Pre-Clinical Data/Mechanism of Action	Serial NCT Number
**Venetoclax (BCL2 inhibitor without BCL XL activity)**					
A Study of Venetoclax and Dexamethasone compared with Pomalidomide and Dexamethasone in subjects with relapsed or refractory Multiple Myeloma	Multiple myeloma	III	Pomalidomide Dexamethasone	Baseline multiple myeloma therapy Bcl-2 one of the critical mechanisms of resistance	NCT03539744
Duvelisib and Venetoclax in Relapsed or Refractory CLL or SLL	Chronic/small lymphocytic leukemia	I/II	Duvelisib	Pre-clinical study showed the combination with duvelisib, a PI3K-δ and PI3K-γ inhibitor reversing the resistance to Bcl2 inhibition (which one to pick)	NCT03534323
Venetoclax in treating participants with recurrent or refractory Mature T-Cell Lymphoma	T cell lymphoma	II	Single agent	Bcl-2 overexpression as a resistance mechanism to conventional therapy in mantle cell lymphma	NCT03534180
Venetoclax, Lenalidomide and Rituximab in patients with previously untreated Mantle Cell Lymphoma	Mantle cell lymphoma	I	Lenalidomide and rituximab	Combination of venetoclax to preclude resistance to standard of care, as first line therapy	NCT03523975
Venetoclax and Ibrutinib in treating in participants with Chronic Lymphocytic Leukemia and Ibrutinib resistance mutations	Chronic lymphocytic leukemia	II	ibrutinib	Overcome the resistant to ibrutinib by combination with venetoclax	NCT03513562
Venetoclax, Lenalidomide and Rituximab in patients with relapsed/refractory Mantle Cell Lymphoma	Mantle cell lymphoma	I/II	Lenalidomide and rituximab	Combination of venetoclax to overcome resistance to standard therapy	NCT03505944
Venetoclax and Vincristine Liposomal in treating patients with relapsed or refractory T-cell or B-cell Acute Lymphoblastic Leukemia	Acute lymphoblastic leukemia	Ib/II	Liposomal vincristine	Addition of venetoclax to the previously shown efficacy of vincristine to induce Philadelphia chromosome negative ALL	NCT03504644
Study of Venetoclax, a BCL2 antagonist, for patients with Blastic Plasmacytoid Dendritic Cell Neoplasm (BPDCN)	BPDCN	I	Single agent	BPDCN depends on BCL2 thus sensitive to venetoclax [203]	NCT03485547
A Study of Venetoclax and Dinaciclib (MK7965) in patients with relapsed/refractory Acute Myeloid Leukemia	Acute myeloid leukemia	II	Dinaciclib	Pre-clinical study showed that CDK9 inhibition mediates venetoclax sensitization	NCT03484520
Study of Venetoclax with the mIDH1 Inhibitor Ivosidenib (AG120) in IDH1-mutated hematologic malignancies	Hematological malignancies with IDH1 mutations	Phase I with dose expansion	Ivosidenib (inhibitor of mutated IDH1)	Preclinical study showing hematological malignancies with mutated IDH1	NCT03471260
**Navitoclax (BCL2, BCL-Xl, BCLw)**					
Navitoclax and Vistusertib in treating patients with relapsed Small Cell Lung Cancer and other solid tumors	Small cell lung cancer, other metastatic solid tumors	I/II	vistusertib	Apoptosis and mTOR inhibition synergistically inhibit the growth of lung cancer (however, the preclinical study was more in NSCLC) [204]	NCT03366103
Osimertinib and Navitoclax in treating patients with EGFR- positive previously treated advanced or metastatic Non-Small Cell Lung Cancer	Non-small cell lung cancer	Ib	Osimertinib (T790M cell sensitive EGFR inhibitor)	Bcl-2 inhibitor navitoclax induce tumor growth inhibition in osimertinib resistant NSCLC [205]	NCT02520778
Navitoclax and Sorafenib Tosylate in treating patients with relapsed or refractory solid tumors	Metastatic solid tumors	I	Sorafenib	BCL-2 proteins determine sorafenib/regorafenib resistance and BH3-mimetic efficacy in hepatocellular carcinoma [206]	NCT02143401
Trametinib and Navitoclax in treating patients with advanced or metastatic solid tumors	Metastatic solid tumors	Ib/II	Trametinib	BCL-2 confers de novo resistance in BRCA mutated solid cancer [207]	NCT02079740
Dabrafenib, Trametinib, and Navitoclax in treating patients with BRAF mutant Melanoma or solid tumors	Solid tumors	I/II	Dabrafenib, trametinib	Combination of RAF/MEK pathway inhibitor along with BCL2 inhibitor, often noted as a resistance mechanism in BRAF mutant solid tumors	NCT01989585

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
