# Peer review of "Novel Apoptosis-Inducing Agents for the Treatment of Cancer, a New Arsenal in the Toolbox"

_cancers, 2019, doi:10.3390/cancers11081087_

Round 1

Reviewer 1 Report

The review manuscript by Lim et al. is extensive and clearly organized, and I did not find any major fault. Therefore, I would recommend this article for the publication in Cancers after the modification of one very minor point.

MINOR POINT

In Figure1, the word "C-FLIP" near the mitochondrion is not necessary. C-FLIP works only in the extrinsic apoptosis pathway, and  the word "FLIP" is already described near the TRAIL-DR4/5 complex.

Author Response

R1: In Figure1, the word "C-FLIP" near the mitochondrion is not necessary. C-FLIP works only in the extrinsic apoptosis pathway, and the word "FLIP" is already described near the TRAIL-DR4/5 complex.

Thank you. We have removed the C-FLIP from figure 1, and also revised the name of FLIP to C-FLIP correctly

Reviewer 2 Report

In this review the authors make an overview on the currently available apoptosis targeting agents used in therapy. With this review they underlined the importance of the rstoration of apoptosis pathway, silenced in many pathologies.

I suggest a minor revision:

1)  the authors could mention (especially in the introduction section) that apoptosis is only one of the types of programmed cell death (PCD) characterized today. A small introduction would improve the already good quality of the review.

2) the figure 2 and 3 are not mentioned in the text

3) the figure 4 is not subdivided in different items, but in the text they refer to figure 4(b-c and d)

4) the figure 4 is not well explained, is it incomplete?

Author Response

R2: In this review the authors make an overview on the currently available apoptosis targeting agents used in therapy. With this review they underlined the importance of the restoration of apoptosis pathway, silenced in many pathologies.

I suggest a minor revision:

1)    the authors could mention (especially in the introduction section) that apoptosis is only one of the types of programmed cell death (PCD) characterized today. A small introduction would improve the already good quality of the review.

Thank you for your insightful comment. We have reflected your suggestion, and added several sentences in the introduction paragraph to discuss other PCD pathways including 10 major cell death proposed by Nomenclature Committee on Cell Death (NCCD) in 2018.

2)    the figure 2 and 3 are not mentioned in the text

Thank you for pointing this out. We looked through the figure comments within the manuscript – and realized the figures that have been changed from the original manuscript version was not reflected back in the manuscript. We have re-organized our figures and the paragraph to reflect current figures.

3)    the figure 4 is not subdivided in different items, but in the text they refer to figure 4(b-c and d)

Yes, this is in line with above issue that now we have addressed.

4)    the figure 4 is not well explained, is it incomplete?

We have added the explanation of figure 4 in the same paragraph explaining the strategies to improve direct anti-apoptosis targeted agent as effective therapeutics. Thank you again for letting us to pay attention on this important gap

Reviewer 3 Report

This review by Dr. Lim and colleagues provides a discussion on the apoptosis-targeting mechanisms and selective agents for cancer treatment. Authors emphasize the central roles of the two major apoptosis-signaling pathways in regulating sensitivity of many drugs and, importantly, conferring resistance to current chemo and targeted therapeutics. With this rationale, authors detail a number of strategies that particularly focus on targeting the extrinsic and intrinsic pathways. In addition to a detailed discussion of various current and emerging approaches to targeting apoptosis signaling, authors highlight significant challenges due to diverse resistance mechanisms and existence of cross-talk pathways. Authors also highlight a number of clinical trials to evaluate clinical suitability and development of various apoptosis-targeting molecules (Biologics, peptides, or small molecules) as well as a need for finding predictive biomarkers to optimize efficacy of such novel therapeutic agents. This well-written review has a good level of relevant details and deserving of publication in Cancers. My minor comments/suggestions are as below:

1.       In line 41, authors need to indicate upfront that TRAIL was not found suitable and the discussion below further highlights the various aspects.

2.       In line 47, (MOMP) that results in cytochrome c release, and apoptosme formation, leading…

3.       Please modify line 50 to state that for the focus of this paper, the agents targeting HSPs and ER stress apoptosis will not be discussed here, and such discussion could be found elsewhere (Refs: please indicate any review/research paper(s) that discuss targeting of HSPs and ER stress apoptosis pathways for development of anti-cancer modalities).

4.       Line 55, advances in genomic asays……………offer critical technologies to move

5.       Line 60, include cIAPs,

6.       Line 62, biology, continues to offer intriguing ………………………….Further efforts to develop………..mechanism therefore are rational and necessary.

7.       Please keep the formatting of reference indication in the text consistent. The majority of the text has references indicated in parentheses however in some places the ref #s are indicated as superscripts.

8.       Line 103, DR-targeting biliogics such as DR4-specific ……………………..

9.       Line 213apoptosis inducer [137, 138].

10.   In line 235, at the end of paragraph please state that all these studies are pre-clinical.

11.   NF-kB, please use kappa symbol throughout.

12.   4.3.2. and 4.6.2. Resiatance Mechanisms

13.    Line 478, HDAC inhibitors induce……..

14.   Line 580, identifying the fraction….

15.   In figure 4, indication of panels A, B, C, and d is missing in the figure.

Author Response

R3: This review by Dr. Lim and colleagues provides a discussion on the apoptosis-targeting mechanisms and selective agents for cancer treatment. Authors emphasize the central roles of the two major apoptosis-signaling pathways in regulating sensitivity of many drugs and, importantly, conferring resistance to current chemo and targeted therapeutics. With this rationale, authors detail a number of strategies that particularly focus on targeting the extrinsic and intrinsic pathways. In addition to a detailed discussion of various current and emerging approaches to targeting apoptosis signaling, authors highlight significant challenges due to diverse resistance mechanisms and existence of cross-talk pathways. Authors also highlight a number of clinical trials to evaluate clinical suitability and development of various apoptosis-targeting molecules (Biologics, peptides, or small molecules) as well as a need for finding predictive biomarkers to optimize efficacy of such novel therapeutic agents. This well-written review has a good level of relevant details and deserving of publication in Cancers. My minor comments/suggestions are as below:

1.       In line 41, authors need to indicate upfront that TRAIL was not found suitable and the discussion below further highlights the various aspects.

Thank you for your suggestion. We have added the sentence: ‘However, TRIAL and DR agonists also have shown several challenges in therapeutic development as we will highlight below (Line 41 in revised manuscrip.’ So the readers can anticipate to learn more about these challenges.

2.       In line 47, (MOMP) that results in cytochrome c release, and apoptosome formation, leading…

Thank you for great recommendation. We have revised the sentence accordingly.

3.       Please modify line 50 to state that for the focus of this paper, the agents targeting HSPs and ER stress apoptosis will not be discussed here, and such discussion could be found elsewhere (Refs: please indicate any review/research paper(s) that discuss targeting of HSPs and ER stress apoptosis pathways for development of anti-cancer modalities).

Thank you for a great suggestion. We cited this paper for readers interest for HSP and ER stress and cell death at the end of introduction part – two review papers for each topic.

4.       Line 55, advances in genomic assays……………offer critical technologies to move

Thank you for great recommendation. We have revised the sentence accordingly (new Line #61-2).

5.       Line 60, include cIAPs,

Thank you for great recommendation. We have revised the sentence accordingly (new Line #66).

6.       Line 62, biology, continues to offer intriguing ……………………Further efforts to develop………..mechanism therefore are rational and necessary.

Thank you for great recommendation. We have revised the sentence accordingly (new Line #68).

7.       Please keep the formatting of reference indication in the text consistent. The majority of the text has references indicated in parentheses however in some places the ref #s are indicated as superscripts.

Thank you. We made sure all reference format is consistent.

8.       Line 103, DR-targeting biologics such as DR4-specific ………………..

Thank you for great recommendation. We have revised the sentence accordingly (new Line #185).

9.       Line 213 apoptosis inducer [137, 138].

Thank you for great recommendation. We have revised the sentence accordingly (new Line #356).

10.   In line 235, at the end of paragraph please state that all these studies are pre-clinical.

Thank you for great recommendation. We have revised the sentence accordingly (new Line #378).

11.   NF-kB, please use kappa symbol throughout.

Thank you for great recommendation. We have revised the sentence accordingly.

12.   4.3.2. and 4.6.2. Resistance Mechanisms

Thank you for great recommendation. We have revised the sentence accordingly.

13.    Line 478, HDAC inhibitors induce….

Thank you for great recommendation. We have revised the sentence accordingly (new Line #627).

14.   Line 580, identifying the fraction….

Thank you for great recommendation. We have revised the sentence accordingly (new Line #753).

15.   In figure 4, indication of panels A, B, C, and d is missing in the figure.

Yes, thank you – another reviewer noticed the same issue and we have addressed this by fixing both the order of figure, and the parag

Reviewer 4 Report

Dear Authors, 

please  amend your review as follows.

Major : 

#1 : The TRAIL or PARAs (pro-apoptotic receptor activators) list is not complete. Authors need to add the CPT, circularly permuted TRAIL developed by Sunbio Biotec (US9289468B2).

#2 : Lines 89-93, Authors are probably aware that there is still a discussion whether DR4 or DR5 are responsible of this pro-metastatic signal transduction (See : Wu S et al. Int J Cancer 2018 PMID: 29446085; Dufour et al. Oncotarget 2017 PMID: 28039489; Oh YT et al. Oncotarget 2015 PMID: 26510914 ), and thus they should probably mention and discuss this issu, in the light of publications highlighting the anti- or pro-metastatic capabilities of TRAIL or PARAs (ie: Lee HY et al. Mol Cancer Res PMID: 30266755; Pal S et al. Cancer Microenviron. PMID: 27106232; Rossini A. et al. Breast Cancer res. PMID: 23053664 or Grosse-Wilde A. et al. JCI PMID: 18079967).

#3 : The Authors should also include the HexaBody®-DR5/DR5 (GEN1029) in their list of PARAs.

#4 : The Chapter related to the development of biomarkers should include TRAIL decoy receptors (DcR1 and DcR2); cFLIP which is a major inhibitor of caspase-8 as well as considerations about DR4 NGlycosylation, in addition to the O-Glycosylation of DR5.

Minor :

Figure #1 Please remove the c-FLIP close to the mitochondria or explain why you have it there. Please change FLIP by c-FLIP in the DISC. 

Author Response

R4: Dear Authors, 

please amend your review as follows.

Major: 

#1 : The TRAIL or PARAs (pro-apoptotic receptor activators) list is not complete. Authors need to add the CPT, circularly permuted TRAIL developed by Sunbio Biotec (US9289468B2).

Thank you for comments. We updated Table 1 to include CPT and related clinical trial information. We also added CPT to the main text as a 2nd generation DR targeting therapy.

#2 : Lines 89-93, Authors are probably aware that there is still a discussion whether DR4 or DR5 are responsible of this pro-metastatic signal transduction (See : Wu S et al. Int J Cancer 2018 PMID: 29446085; Dufour et al. Oncotarget 2017 PMID: 28039489; Oh YT et al. Oncotarget 2015 PMID: 26510914 ), and thus they should probably mention and discuss this issu, in the light of publications highlighting the anti- or pro-metastatic capabilities of TRAIL or PARAs (ie: Lee HY et al. Mol Cancer Res PMID: 30266755; Pal S et al. Cancer Microenviron. PMID: 27106232; Rossini A. et al. Breast Cancer res. PMID: 23053664 or Grosse-Wilde A. et al. JCI PMID: 18079967).

We have expanded the discussion of both the anti-cancer effects and pro-metastatic effects of DR4 and DR5. Our read of the literature including those cited above by the reviewer, suggest that both DR4 and DR5 can induce apoptosis and which predominates depends on the tumor type being studied.  The pro-metastatic effects so far are not well elucidated and the only paper that directly compared DR4 and DR5 (Dufour et al, Oncotarget 2017) found that DR4 induced apoptosis while DR5 induced motility and invasion.

#3 : The Authors should also include the HexaBody®-DR5/DR5 (GEN1029) in their list of PARAs.

We updated Table 1 now includes Hexabody. It is also added to the main text.

#4 : The Chapter related to the development of biomarkers should include TRAIL decoy receptors (DcR1 and DcR2); cFLIP which is a major inhibitor of caspase-8 as well as considerations about DR4 NGlycosylation, in addition to the O-Glycosylation of DR5.

A discussion of N-glycosylation of DR4, the role of Decoy Receptors, and the role of c-FLIP are now added to the chapter that describes various biomarkers to predict TRAIL sensitivity.

Minor:

Figure #1 Please remove the c-FLIP close to the mitochondria or explain why you have it there. Please change FLIP by c-FLIP in the DISC. 

·       This is overlapping comment with the one from reviewer #1. We have corrected error in the figure 1. Thank you.

Round 2

Reviewer 4 Report

The revise version is fine for me.